# Unraveling the ARC Puzzle: Mimicking Human Solutions with Object-Centric Decision Transformer

**Jaehyun Park   Jaegyun Im   Sanha Hwang   Mintaek Lim   Sabina Ualibekova   Sejin Kim   Sundong Kim** [1]

## Abstract

In the pursuit of artificial general intelligence (AGI), we tackle Abstraction and Reasoning Corpus (ARC) tasks using a novel two-pronged approach. We employ the Decision Transformer in an imitation learning paradigm to model human problem-solving, and introduce an object detection algorithm, the Push and Pull clustering method. This dual strategy enhances AI's ARC problem-solving skills and provides insights for AGI progression. Yet, our work reveals the need for advanced data collection tools, robust training datasets, and refined model structures. This study highlights potential improvements for Decision Transformers and propels future AGI research.

## 1. Introduction

With the advent of deep learning, AI models have begun to outperform humans in various tasks. However, these models still have limitations in their adaptability, especially when dealing with unforeseen situations (Borji, 2023). To overcome this hurdle, researchers are working to imbue AI with abstraction and reasoning skills, attempting to teach machines to think like humans (Chollet, 2019). Such abilities include inferring new knowledge from what they have already learned and flexibly responding to novel situations.

A key benchmark dataset for evaluating these abilities, the Abstraction and Reasoning Corpus (ARC), was proposed by Francois Chollet (Chollet, 2019). The ARC comprises 2-5 input-output pairs conforming to a specific rule, grounded in a variety of concepts such as object relations, numbers, symmetry, and quantification. While humans can readily solve these problems, existing machine-learning solutions strug-

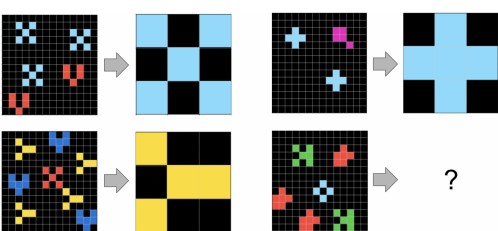

*Figure 1.* An example of an ARC problem. The model should be able to look at 3-5 input-output grid pairs and predict the correct output answer grid when given the final input grid.

gle, with the highest accuracy in ongoing ARC competitions standing at around 30 percent.[1][2]

Existing high-performing solutions for ARC problems typically rely on hard-coding or random search methods (Ainooson et al., 2023). To overcome these limitations, several researchers have attempted to reduce the search space using domain-specific languages (DSL) (Ellis et al., 2021; Banburski et al., 2020), graph structures (Xu et al., 2023), or even natural language processing (Acquaviva et al., 2022).

In this study, we follow an imitation learning approach, a sequential task requiring learners to emulate expert behavior for optimal performance (Attia & Dayan, 2018). This approach is based on the hypothesis that modeling human thinking processes can help bridge the performance gap between machine learning models and human-level intelligence (Johnson et al., 2021; Lake et al., 2017). Previous studies have pointed out that many machine learning models struggle with reasoning beyond their existing knowledge, which hampers their progress towards human-like intelligence (Lake et al., 2017).

In response to these findings, we set out to collect human problem-solving traces using the Object-Oriented ARC (O2ARC) interface (Kim et al., 2022). With this valuable dataset on hand, we chose to leverage the Decision Transformer (Chen et al., 2021), a method known for its effectiveness in tasks similar to ours, for our imitation learning

[1]AI Graduate School, GIST, South Korea. Correspondence to: Jaehyun Park <jaehyun00518@gmail.com>, Sundong Kim <sdkim0211@gmail.com>.

*Interactive Learning with Implicit Human Feedback Workshop* at ICML 2023. Copyright 2023 by the author(s).

---

[1]For ongoing ARC competition details, visit https://lab42.global/arcathon/.

[2]For the essay challenge, see https://lab42.global/past-challenges/essay-intelligence/.

approach. By building on the problem-solving strategies captured in the O2ARC dataset, we aim to enhance the machine's ability to reason and solve ARC problems more effectively.

In addition to learning from human strategies, we also observed that humans often perceive ARC problems in terms of objects. Inspired by this observation, we propose a new object detection method, the Push and Pull (PnP) clustering algorithm. Tailored specifically for ARC tasks, this algorithm detects and understands objects within a problem, channeling this information into the Decision Transformer to enhance the accuracy of its predictions.

Our combined approach, which we dub the 'Object-centric Decision Transformer', has led to significant improvements in solving ARC problems. It has shown promising results on four representative problems: *diagonal flip*, *tetris*, *gravity*, and *stretch*. We believe that this approach charts a potential pathway toward machines that are capable of solving abstract reasoning tasks more effectively.

## 2. Related Work

### 2.1. Mini-ARC and O2ARC Tool

The Abstraction and Reasoning Corpus (ARC) problems cover a broad size range, from $1\times1$ to $30\times30$ grids. This variability significantly affects the solution process, making model generalization across different sizes challenging. To mitigate this, the Mini-ARC dataset (Kim et al., 2022) was introduced, providing fixed-size problems where both input and output grids are $5\times5$. This uniformity simplifies the learning task and streamlines model training and evaluation. The Mini-ARC dataset, comprising 150 examples, follows the ARC dataset's structure. To complement the dataset, the O2ARC tool was introduced to capture human problem-solving processes. This tool gathers vital data, including problem-solving time, the domain-specific language (DSL) used, and the number of attempts. The O2ARC tool provides insights into human problem-solving strategies on Mini-ARC tasks, valuable information for enhancing AI models.

### 2.2. Human-like problem solving in AI

Various strategies exist for developing AI that can emulate human-like behavior. One such approach relies on self-training the model on minimal data (Lake et al., 2017), as exemplified by systems like Dreamcoder (Ellis et al., 2021). In contrast, a second approach uses expansive models and large amounts of data to learn intelligent behaviors, a method typified by Large Language Models (LLMs) using transformers (Brown et al., 2020). For our exploration of human-like problem-solving, we favor this second approach. Specifically, we are interested in the application of the transformer structure used in LLMs to facilitate learn-ing that emulates human behavior (Melo, 2022). Building upon this concept, we focus on offline reinforcement learning : Decision Transformer (Chen et al., 2021) (DT) and Behavior Cloning (Torabi et al., 2018) (BC) . As a ground-breaking approach in offline reinforcement learning, DT offers unique insights into the development of AI capable of human-like problem-solving. DT trains policies based on existing data, using a transformer model for conditional sequence modeling. Similarly, BC is renowned for its ability to effectively mimic human behavior. Consequently, we propose a compelling question: could supervised learning techniques that mimic human behavior, using these tools, also solve complex datasets such as the ARC? This question motivates our proposed investigation into creating an AI that emulates human behavior.

### 2.3. Object Detection in ARC Problems

Object detection, a pivotal element of image recognition and understanding, has seen considerable research progress and impressive performance gains. Various models (Ren et al., 2015; Redmon et al., 2016; Carion et al., 2020; Chen et al., 2017) have advanced the field from simple CNN structures to transformer incorporation. Among these advancements, Slot Attention (Locatello et al., 2020; Singh et al., 2023) emerges as a particularly effective framework for object detection within images. This approach deftly captures object properties and enables accurate reconstruction. Slot Attention represents a significant step forward in this area, successfully addressing complex object detection scenarios and establishing a solid foundation for further exploration and development. Depite the promise of these models, it is crucial to note that they have been primarily designed for traditional computer vision tasks and are not specifically tailored for ARC problems. Since ARC problems comprise 2D arrays of small, single-channel grids (in most cases smaller than the MNIST dataset), traditional CNN architectures may struggle to capture their properties effectively (Lin et al., 2017). Transforming these 2D arrays into a 3D-channel representation via image super-resolution (Kim et al., 2016) could potentially help, but this may not fully account for ARC problems' inherent abstract patterns and small grid sizes. Hence, we propose the Push and Pull (PnP) clustering algorithm, designed to effectively detect objects in ARC problems by directly addressing these challenges.

## 3. Method

### 3.1. Imitation Learning: Decision Transformer

#### 3.1.1. ARCHITECTURE

Humans leverage their accumulated knowledge to solve problems. To address specific tasks, they come up with DSLs by scrutinizing each example, integrating them, and

devising a preliminary solution. If a particular combination of DSLs results in the correct solution, they apply it to other tasks to validate the effectiveness of the approach (Nye et al., 2020). Therefore, we want to encourage the computer to solve the ARC problem using a similar cognitive process. To this end, we adapted the Decision Transformer, which we identified as the most appropriate model for imitating human learning.

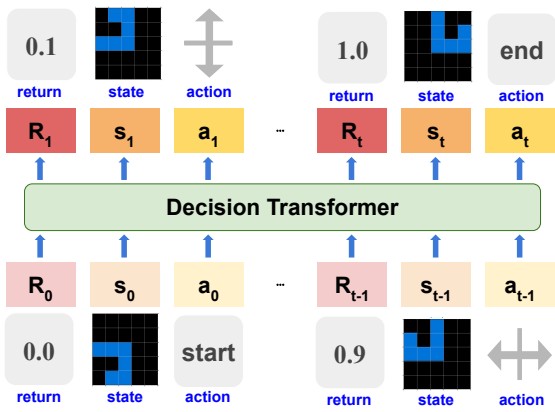

*Figure 2.* Training a Decision Transformer with Mini-ARC trace. A Decision Transformer utilizes the return-to-go, state, and action at time $t$ as input and generates a prediction of the following time step, $t+1$.

The Decision Transformer (Chen et al., 2021) operates on three distinct inputs per time step: return-to-go, state, and action. The return-to-go is calculated by initiating the state at 0, designating the final state as 1, and partitioning the interval between these states into equal segments. The state is represented using a 5×5 input grid, same with ARC problems. We used 14 actions provided by the O2ARC tool, such as clockwise rotation, coloring, left-right flip, and up-down flip. Each of these three inputs—return-to-go, state, and action— are converted to embedding vectors through their corresponding embedding layers.

In the ARC problem, discerning the relationship between individual pixels is important, so the state is processed as a pixel-by-pixel embedding vector akin to ViT (Dosovitskiy et al., 2021) approach, rather than being transformed by a single, unified embedding vector. Figure 2 shows the structure of the Decision Transformer (DT) model adapted on our tasks. Based on the provided inputs, the model predicts the state, action, and return-to-go for the subsequent time step, which allows us to project the entire problem-solving process.

### 3.1.2. DATA AUGMENTATION

Offline reinforcement learning has the potential to achieve impressive results when provided with a sufficient amount of high-quality data about the environment. Therefore, data augmentation is an essential consideration. However, conventional data augmentation techniques can be challenging to implement for ARC due to the limited number of input-output pairs. To overcome this limitation, the expert traces of human solution processes were applied to randomly generated grids, which were utilized as a training dataset. In more detail, to collect the human solution processes, the O2ARC Tool was utilized (Kim et al., 2022).[3] Through this O2ARC Tool, the solutions provided by human participants were collected. These solutions were then manually examined to extract expert traces. For each randomly generated grid, the expert trace was applied and each step is collected. Employing this approach, we generated 10,000 training instances and 2,000 testing instances per task. Figure 3 demonstrates the overall procedures for our data augmentation process.

### 3.2. Object Detection: PnP Clustering Algorithm

Drawing inspiration from previous work (Xu et al., 2023), we concentrated on developing an object detection approach for the ARC task.[4] Our method entails representing ARC problems as graph abstractions with nodes embodying co-ordinates, colors, position index, and distance (weighted edge) illustrating the relationship between the two connected nodes. In this graph representation, each pixel on the grid becomes a graph node, with the distance varying based on the relationship between the two connected nodes. To determine edge weights, we investigated all 400 ARC problems and categorized them. Through this process, we identified 128 problems that could be defined as object-centric. This allowed us to discern objects that humans could heuristically recognize, which possessed relationships between each pixel. The edge between nodes signifies the 'relative distance' between them, with closer nodes marked by a small distance value. We employed the concept to determine the distance values for the edges. More details are outlined in Table 1 and Appendix C. The Push and Pull Clustering Algorithm (PnP algorithm) consists of three main steps: 1) the abstraction function $f$, 2) the clustering function $g$, and 3) DBSCAN (Ester et al., 1996) for detecting clusters.

#### 3.2.1. GRAPH ABSTRACTION

The abstraction function $f$ takes the original grid as an input and provides corresponding graph abstraction. This abstraction comprises a list of node objects and an adjacent matrix to represent the edge information. Each individual pixel corresponds to a single node in the graph, with color, coordi-

---

[3]An ARC trace refers to the sequence of steps used in solving an ARC problem.

[4]Previous studies explored ARC solutions using the concept of objects, but only two definitions are used to detect simple objects.

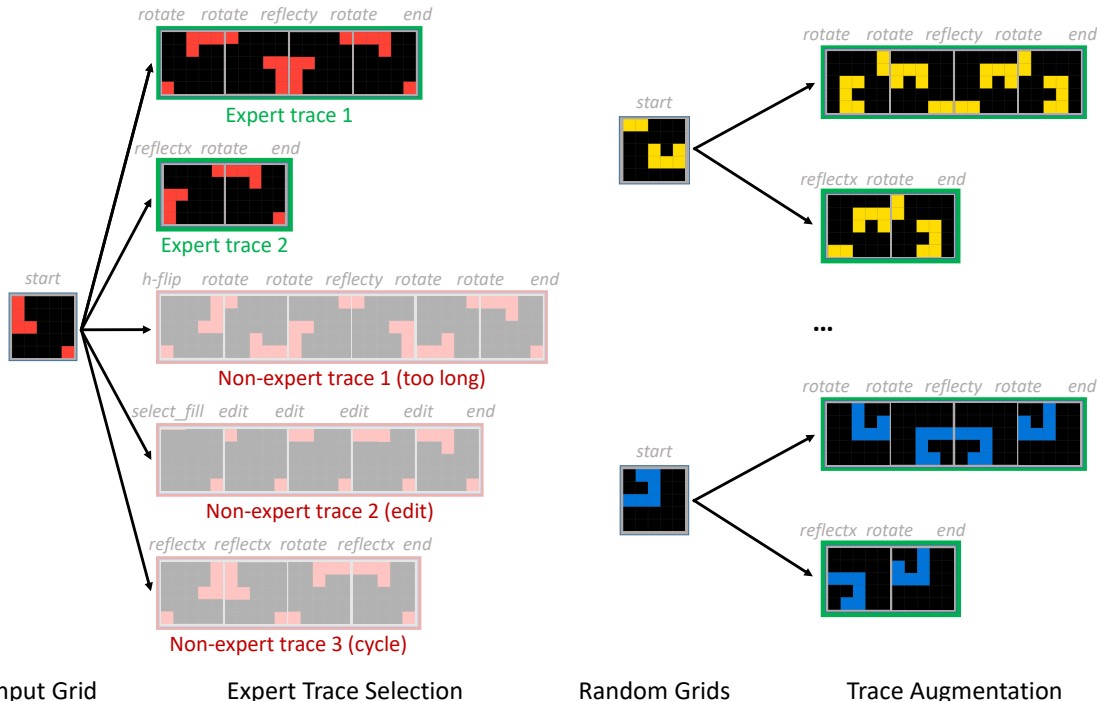

*Figure 3.* Trace augmentation process for the *diagonal flip* problem: Firstly, from the human solution processes for a particular input grid of the *diagonal flip* problem, we select the expert traces. To be classified as an expert trace, the length of the trace must not be too long, the `edit` operation should not be used, and there must be no cycles in the solution process. The selected expert traces are then applied identically to each randomly generated grid, which is how we perform trace augmentation.

nate, and index properties. The color attribute is originated from the color of the corresponding pixel in the array. Every black pixel is considered as a background and assigned a color value of 0. The coordinate information is assigned in a sequential manner by tripling the index of the original array. Two nodes are connected by an edge if they share sides or vertices, indicating direct or diagonal adjacency. Each edge's distance is determined by the relationship between them, as detailed in Table 1.

*Table 1.* Relative distance between nodes. Greater distance reduces the likelihood of the nodes being part of the same object.

| Category | distance |
|---|---|
| Same color, Direct adjacency | 1 |
| Same color, Diagonal adjacency | 2 |
| Different color, Direct adjacency | 4 |
| Different color, Diagonal adjacency | 5 |

#### 3.2.2. PUSH AND PULL OPERATION

The graph is then subjected to a push and pull operation for clustering, which we denote as function $g$. The operation adheres to Equation (1), using the distance values outlined in Table 1. As mentioned before, edge distances signify the

relative proximity of adjacent nodes.

$$\text{Operation} = \begin{cases} \text{Push} & \text{if } \frac{\text{Distance of Edge}-3}{2} > 0, \\ \text{Pull} & \text{else} \end{cases} \quad (1)$$

If the result derived from Equation (1) is negative, it implies a strong relationship between the two nodes, suggesting they're likely part of the same object, and thereby, a pull operation is triggered. In contrast, if the value of Equation (1) is positive, it signals a weaker relationship between the nodes, prompting a push operation. In both cases, a larger absolute value results in a stronger push or pull action. These calculated values guide the push and pull operations, which modify the coordinates of the nodes in the direction of the connected edge, reflecting the appropriate push or pull action. Applied to every edge, this operation adjusts the coordinate properties of connected nodes. As a result, the output of the clustering function $g$ is a modified graph containing clustered nodes, as depicted in Figure 5.

#### 3.2.3. ASSIGNING CLUSTER TO PIXELS

The final step in our process involves object detection using a conventional clustering algorithm. The output of the push and pull operation $g$ is a graph, which we need to detect objects and assign specific clusters to each one. The outcome

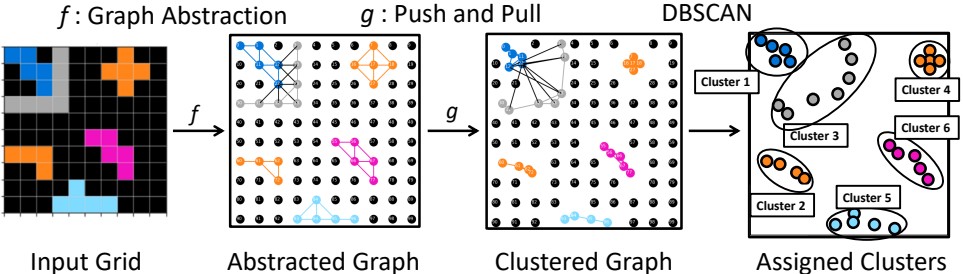

Figure 4. Demonstration of the PnP Clustering Algorithm. Function $f$ abstracts the ARC problem into a graph, then function $g$ applies the push and pull operation to form object clusters. The PnP algorithm enables effective object detection while being computationally efficient compared to other methods.

of this step should provide information about which pixels constitute an object. To achieve this, we employed a density-based model known as DBSCAN (Ester et al., 1996), which groups nodes based on their distance measurements. Since the nodes identified as part of the same object by the PnP algorithm are clustered, DBSCAN finalizes them into single groups. Unlike other clustering models, such as k-means, DBSCAN does not require a predetermined number of clusters, making it more adaptable and effective for our task. That's the reason that we adapted DBSCAN for assigning clusters to each pixel.

### 3.3. Integration of Decision Transformer and Object detection

While applying the Decision Transformer to the ARC problem successfully predicts action and return-to-go, it tends to make errors when predicting states, often inaccurately identifying changes in pixels. To improve accuracy, we augment the model with additional object information, enabling it to more precisely predict each pixel. Prior studies, such as the Prompt Decision Transformer (Xu et al., 2022), demonstrated that performance is improved when we use Decision Transformer with additional information including basic elements like return-to-go, state, and action. So we decided to give object information that objects identified by the PnP algorithm were inputted into the Decision Transformer in the same manner as the state, action, and return-to-go. Consequently, we incorporated the PnP algorithm to provide object information into the Decision Transformer for the ARC problem. Figure 5 shows the operation of the embedding in the Decision Transformer with the PnP algorithm.

## 4. Experiments

### 4.1. Experimental Setting

#### 4.1.1. TRACE AUGMENTATION

Training the Decision Transformer (DT) model presents a unique challenge, as it calls for a considerable volume of training examples complete with traces. However, in the context of the ARC problem, we have traces for just a single test input-output pair. To overcome this constraint, we undertook a strategy of data augmentation. This involved the generation of randomized input grids, but importantly, it maintained the human solution process in each case. Thus, we were able to create a broader range of examples for model training while keeping the essence of the solution strategies intact.

Even though each task in our dataset follows the same fundamental logic, there can be multiple ways to solve with provided DSLs. To capture this variation, we gather 3-10 distinct solution traces per task, enriching our training set. Paired with the randomly generated input grids, these solution guide create a broad array of augmented ARC traces. Every solution process is comprehensively captured in a JSON file, forming a dataset rich in diversity.[5] Figure 4 offers an illustrative example, showcasing augmented ARC traces for our target task: the *diagonal flip*.

#### 4.1.2. DECISION TRANSFORMER

The training dataset includes 10,000 complete solution traces per task, while the evaluation dataset comprises 2,000 input-output pairs. For training, each solution trace is truncated to a maximum of five time steps. To compensate for shorter traces, we pad the data with values identical to the starting grid, ensuring uniform data length. The model employs cross-entropy loss (CE) for both state and action predictions. For Return-to-go predictions, which range between 0 and 1, we use Mean Squared Error (MSE) loss due

---

[5]The appendix describes the JSON format of an ARC trace.

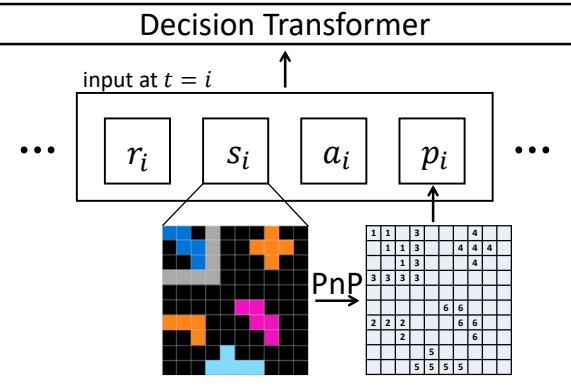

*Figure 5.* Diagram illustrating a method in which the PnP algorithm provides additional information to the Decision Transformer. For every input at $t = i$, we extract current state from $s_i$ and apply PnP algorithm to generate $p_i$. The 2-dimensional grid $p_i$ includes object information.

to its continuous nature. Consequently, the total loss of the Decision Transformer is defined as shown in Equation 2.

$$\ell_{DT} = \sum_{i=t}^{t+R} CE(s_i, \hat{s}_i) + CE(a_i, \hat{a}_i) + MSE(r_i, \hat{r}_i), \quad (2)$$

where the Decision Transformer loss $\ell_{DT}$ is minimized over a range from $t$ to $t + R$. $CE(s_i, \hat{s}_i)$ and $CE(a_i, \hat{a}_i)$ represent the cross-entropy loss between the true and predicted values of $s_i$ (state at time $i$) and $a_i$ (action at time $i$), respectively. $MSE(r_i, \hat{r}_i)$ denote the mean squared error between the true value of $r_i$ (return-to-go at time $i$ and its predicted counterpart.

During the evaluation phase, only the initial input value is provided. The model then generates the next time step's return-to-go, state, and action, iterating this process until an 'end' action is returned. Should the generation process exceed a predefined number of time steps, the model assumes it failed to solve the task and discontinues its predictions. Lastly, we calculate the model's accuracy based on a comparison of its final outputs with the correct answers.

### 4.1.3. PNP ALGORITHM

To enhance the performance of the Decision Transformer model, we incorporate additional object information indicating which pixels belong to the same object. The PnP algorithm functions as a data pre-processing encoder: it ingests 2-dimensional ARC data, detects the objects, and records the object information at each node. This algorithm was tested on 128 object-centric problems and demonstrated a recall of 88 percent. This suggests that the PnP algorithm can accurately detect the presence of objects in the ARC

problem with a success rate of 88 percent. Detailed performance metrics of the PnP algorithm will be presented in section D. Therefore, we propose that if object information can be refined and structured appropriately for inclusion in the Decision Transformer model, it could significantly enhance the model's understanding of object-based tasks.

The PnP algorithm generates raw output in graph form. Each node, classified as part of the same object, stores a unique integer identifier, which signifies its object's property. Nodes corresponding to the black color are considered background and assigned a value of -1. Subsequently, we construct a new 2-dimensional matrix mirroring the current state of input data size. This matrix includes the object properties at each element, corresponding to the pixel locations in the original data. The only modification implemented at this stage involves adjusting the background node values to zero, achieved by adding 1 to the object property values of every node, thus facilitating clearer object recognition. As shown in Figure 5, the utilization of this methodology, which involves extracting the grid from $s_i$ and generating $p_i$ using the PnP algorithm at each time step, presents a valuable meanings of enhancing the Decision Transformer's performance by incorporating supplementary object information.

### 4.2. Results

#### 4.2.1. DECISION TRANSFORMER PERFORMANCE

We evaluated the Decision Transformer's problem-solving capabilities on a dataset comprising 2,000 evaluation instances for each of the *diagonal flip*, *tetris*, *gravity*, and *stretch* problems. Figure 6a, 6b, 6c and 6d respectively illustrate the results obtained for the *diagonal flip*, *tetris*, *gravity*, and *stretch* problems. Interestingly, for a given *diagonal flip* problem, the Decision Transformer devises unique solution strategies, as evidenced by the differing actions employed by the two solvers depicted in Figure 6a. The vanilla Decision Transformer model achieved accuracies of 76.51%, 71.51%, 46.72%, and 69.98% for the *diagonal flip*, *tetris*, *gravity*, and *stretch* respectively on the evaluation dataset.

#### 4.2.2. DECISION-MAKING PROCESS ANALYSIS

By integrating additional object information into the Decision Transformer, we conducted training and evaluation in the same environment. As highlighted in Table 2, the Decision Transformer, when augmented with the PnP algorithm, scored an accuracy of 89.96% for the *diagonal flip* problem and 83.80% for the *tetris* problem. This denotes respective performance improvements of 13.45% and 12.29% over the standalone Decision Transformer. Figure 7 depicts results achieved using the standalone Decision Transformer and the PnP-augmented model respectively. In Figure 7(a), the model struggles to correctly identify the 2x2 orange and

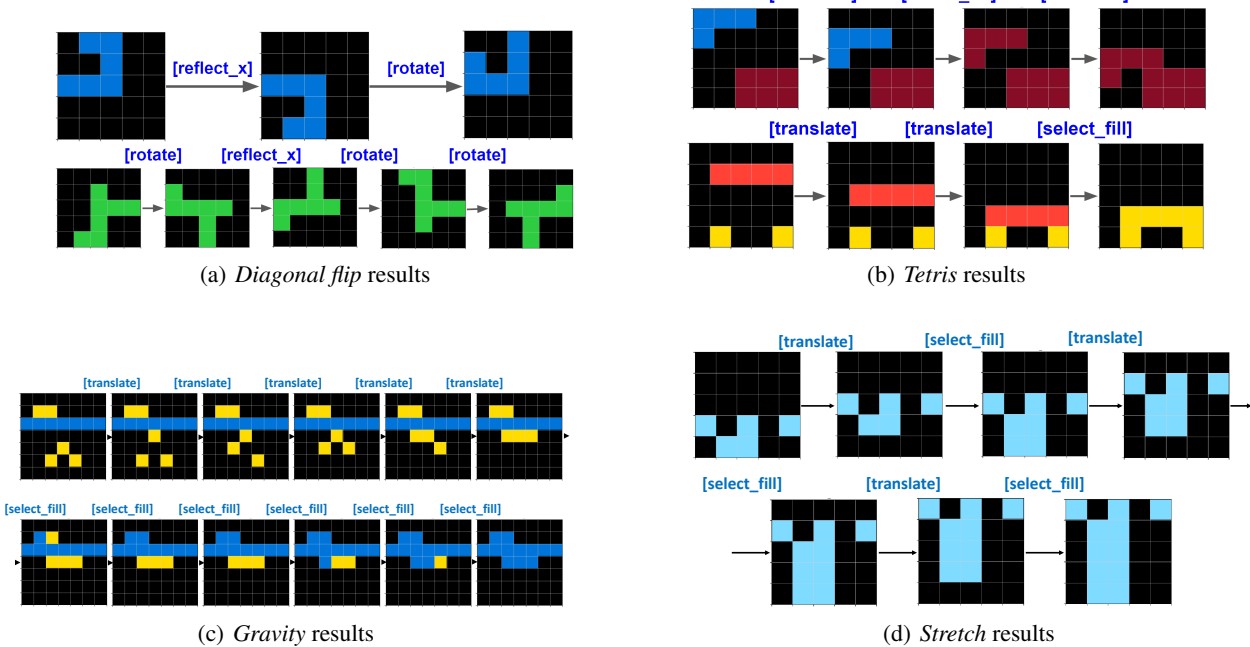

(a) *Diagonal flip* results

(b) *Tetris* results

(c) *Gravity* results

(d) *Stretch* results

*Figure 6.* Comparison of the Decision Transformer's predictions for four different tasks. Decision Transformer learns the patterns and selects the correct action following the established rules.

*Table 2.* Task-wise accuracy of the Decision Transformer and its variations. Noticeable performance improvements are observed when the Decision Transformer is supplemented with additional object information. BC refers to Behavioral Cloning, excluding $r_i$ from the input, while No DT baseline with transformer backbone, does not use any trace information. No DT baseline was not able to solve any test cases.

|  | Diagonal Flip | Tetris | Gravity | Stretch |
|---|---|---|---|---|
| No DT | 0.00 | 0.00 | 0.00 | 0.00 |
| BC | 30.37 ± 0.31 | 80.85 ± 0.61 | 50.01 ± 0.91 | 64.43 ± 1.12 |
| BC (+PnP) | 72.37 ± 0.91 | **91.28** ± 0.47 | **59.15** ± 0.93 | 79.26 ± 0.58 |
| DT | 76.51 ± 0.75 | 71.51 ± 0.66 | 46.72 ± 1.11 | 69.98 ± 1.12 |
| DT (+PnP) | **89.96** ± 0.72 | 83.80 ± 0.47 | 59.00 ± 1.00 | **86.41** ± 0.61 |

pink objects, while in Figure 7(b), these objects are recognized correctly and the resulting movement ensures no overlap with the object at the bottom.

## 5. Discussion

### 5.1. Beyond Traditional Augmentation

In deep learning, numerous data augmentation techniques exist. such as adding noise, cropping, rotation, and color changes, which have proven particularly effective for image datasets (Cubuk et al., 2020). These techniques can be invaluable when data is insufficient or fails to capture representative features. However, applying these common

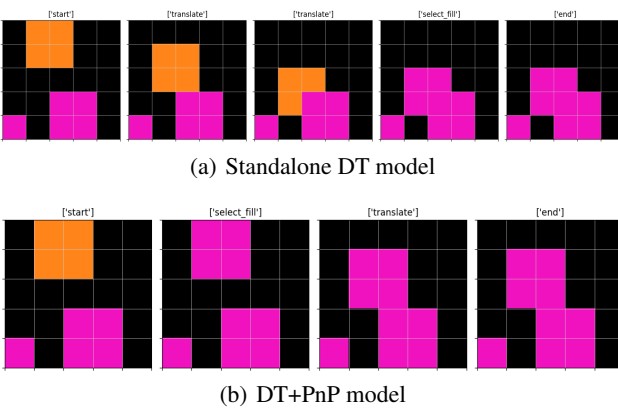

(a) Standalone DT model

(b) DT+PnP model

*Figure 7.* Traces regenerated by Decision Transformer - A comparison of state spaces generated from the standalone DT model, and the DT+PnP model. When augmented with the PnP algorithm, the model identifies and interacts with objects more effectively, avoiding unnecessary downward movement of partial blocks, thereby adhering to the fundamental rules of Tetris.

methods to the ARC (Abstraction and Reasoning Corpus) problem presents unique challenges. Specifically, the nature of ARC tasks, where even slight alterations like color changes or shifts in orientation could drastically alter the problem, renders traditional augmentation techniques less viable. The unique demands of ARC tasks, which require models to infer solutions from limited yet precise exam-

ples, add complexity to the augmentation process. In this study, we confined our data augmentation to methods that do not directly interfere with the problem's inherent properties. Looking ahead, we anticipate that the development of cognition-driven, sequence-preserving augmentation techniques could enhance the learning capacity of the Decision Transformer, enabling it to solve a broader range of problems.

### 5.2. Enhancing Accuracy through Object Detection

The Decision Transformer operates by choosing the optimal action based on the existing trace. However, in the ARC context, it must predict the grid state and action, each affecting the subsequent step. Our analysis revealed that while actions were often predicted correctly, state predictions were frequently erroneous. By specifying objects with the PnP algorithm, state predictions could become more accurate. The performance improvement seems to occur even when the PnP algorithm identifies incorrect objects, likely due to the provision of pixel relationship information.

Experiments demonstrated the tangible impact of object identification on problem-solving efficacy, particularly evident in the *tetris*. This problem used data augmentation to showcase a four-pixel object. Its trace also integrated object-moving actions, emphasizing the importance of object information to the Decision Transformer's performance. In light of this, it's apparent that more nuanced information about the tasks, such as object identities and relationships, could significantly enhance the performance of the solver. This could be achieved through the use of more sophisticated data collection tools, which capture fine-grained user logs. These logs would provide richer and more informative traces for the model, consequently refining the Decision Transformer's accuracy, especially in object-oriented ARC problems.

### 5.3. Limitations of Decision Transformer: The Issue of Dataset Absence

Decision Transformers rely on offline datasets to inform policy training, resulting in a potential adaptability gap when faced with unseen inputs. This limitation was particularly evident in the *tetris*, which required a larger training dataset due to its more complex solving process compared to the *diagonal flip*, leading to diminished performance. Possible remedial strategies could involve the incorporation of approaches such as the Prompt Decision Transformer (Xu et al., 2022), along with a more robust training dataset. With the foundation of recognized patterns through existing training, we anticipate that the system could independently analyze novel ARC tasks, given additional inputs such as prompts (Zhou et al., 2023).

### 5.4. Toward Comprehensive ARC Solutions

Our research highlights the potential of Decision Transformers for some ARC tasks, but also outlines key areas for further investigation. First, we must confirm its effectiveness across varied tasks, by testing its adaptability to ARC tasks with fluctuating lattice sizes. As we handle more tasks, we must carefully review the model's structure and how we train it. At the moment, our paradigm employs a singular model, trained specifically to resolve an individual task. While this strategy proves effective for isolated tasks, it surfaces concerns regarding its suitability for addressing open tasks. One particular challenge lies in designing a competent meta-classifier, which accurately categorizes the task at hand, and channels it to the relevant Decision Transformer. Refining this component is essential for the overall effectiveness of the model. Moving forward, an improved Decision Transformer architecture, incorporating an enhanced meta-classifier, could offer a significant step forward in solving ARC problems. This potential is a focal point of our future work.

## 6. Conclusion

In this study, we aimed to solve ARC problems by emulating human problem-solving strategies, focusing on example observation, DSL selection, and combination to devise solutions. Our employment of the Decision Transformer to replicate human imitation learning demonstrated promising results on the four representative ARC problems, in average of 66.18%, suggesting its applicability to other ARC problems given sufficient data.

Crucially, our approach acknowledges the key role of object recognition and relationship understanding in ARC problem-solving, components integral to approximately half of these tasks. To support this aspect, we proposed a fast Push and Pull (PnP) algorithm, tailored for ARC tasks. The PnP algorithm enhances object identification clarity, bolstering the application of the Decision Transformer. The accuracy significantly improved to an average of 13.61% when we combined the PnP algorithm with the Decision Transformer. We anticipate this object-clarifying advantage of the PnP algorithm will be beneficial across various ARC approaches, potentially enhancing problem-solving strategies.

## Acknowledgement

This work was supported by the IITP (RS-2023-00216011) and the National Research Foundation (RS-2023-00240062) grants funded by the Ministry of Science and ICT, Korea.

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

## A. O2ARC Tool

Our work builds upon the O2ARC tool proposed in the previous study (Kim et al., 2022) for gathering expert human traces. However, we identified certain limitations in the original tool, which led us to suggest and develop several improvements to enhance the collection of human traces that can be used as input for the Decision Transformer. The following points highlight the enhancements made to the original O2ARC tool, drawing inspiration from human cognition:

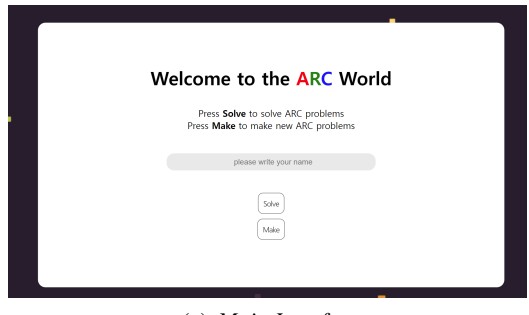

(a) *Main Interface*

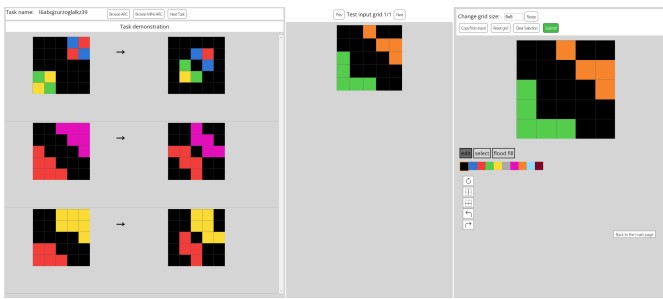

(b) *Solve Interface*

*Figure 8.* O2ARC Tool Interface; you can try to use our O2ARC tool in `https://bit.ly/ARC-GIST`.

- *Addition of Original ARC Problems*: In the original version of O2ARC, only Mini-ARC problems were available. To increase the diversity of problem types, we introduced original ARC problems into the tool. This expansion allows for a wider range of problem-solving scenarios and promotes a more comprehensive understanding of human decision-making processes.

- *Evaluating Traces*: To ensure the quality of collected traces, we introduced a dedicated page within the O2ARC tool for evaluating traces. Traces are categorized as either "good" or "bad" based on their accuracy and potential to disrupt the training process. Evaluators can review and grade the decision-making processes of other users by responding to specific questions posed for each trace. This evaluation process helps in selecting high-quality traces for further analysis and training.

- *Proper operation of the existing functions*: We fixed the existing bugs in the rotation and reflection functions to ensure their proper operation, resulting in better usability and higher-quality traces. Users can now utilize these functions effectively, even outside the given grid, expanding their range of applications.

By incorporating these enhancements, the improved O2ARC tool enables the aggregation of human traces into a single JSON file. This file captures essential information such as the username, task ID, and a sequence of actions performed by users. It provides a detailed account of the tools utilized and the subsequent modifications made to the output grid during the problem-solving process. The following code snippet illustrates the structure of the output JSON file:

```
1  {
2      "id": 1779,
3      "task_id": "Reflection_l6ab2g1dkofxrxht5h",
4      "user_id": "le3k5gb6biqmr9u1nww_ds",
5      "action_sequence": "{"action_sequence": [{"action": {"tool": "start"}, "grid": [[0, 0,
       0, 0, 0], [0, 0, 0, 0, 0], [0, 0, 0, 0, 0], [0, 0, 0, 0, 0], [0, 0, 0, 0, 0]], "
       currentLayer": 0, "layer_list": [[[0, 0, 0, 0, 0], [0, 0, 0, 0, 0], [0, 0, 0, 0, 0],
       [0, 0, 0, 0, 0], [0, 0, 0, 0, 0]]], "submit": 0, "time": 8}, {"action": {"tool": "
       copyFromInput"}, "grid": [["2", 0, "2", 0, "2"], [0, "2", 0, "2", 0], [0, 0, 0, 0, 0],
       [0, 0, 0, 0, 0], [0, 0, 0, 0, 0]], "currentLayer": 0, "layer_list": [[["2", 0, "2",
       0, "2"], [0, "2", 0, "2", 0], [0, 0, 0, 0, 0], [0, 0, 0, 0, 0], [0, 0, 0, 0, 0]]], "
       submit": 0, "time": 3}, {"action": {"tool": "reflectx", "selected_cells": [{"row": 0,
       "col": 0, "val": "2", "selected": true}, {"row": 0, "col": 2, "val": "2", "selected":
       true}, {"row": 0, "col": 4, "val": "2", "selected": true}, {"row": 1, "col": 1, "val":
        "2", "selected": true}, {"row": 1, "col": 3, "val": "2", "selected": true}, {"row": "
       0", "col": "1", "val": "0", "selected": true}, {"row": "0", "col": "3", "val": "0", "
       selected": true}, {"row": "1", "col": "0", "val": "0", "selected": true}, {"row": "1",
```

```
     "col": "2", "val": "0", "selected": true}, {"row": "1", "col": "4", "val": "0", "
  selected": true}, {"row": "2", "col": "0", "val": "0", "selected": true}, {"row": "2",
   "col": "1", "val": "0", "selected": true}, {"row": "2", "col": "2", "val": "0", "
  selected": true}, {"row": "2", "col": "3", "val": "0", "selected": true}, {"row": "2",
   "col": "4", "val": "0", "selected": true}, {"row": "3", "col": "0", "val": "0", "
  selected": true}, {"row": "3", "col": "1", "val": "0", "selected": true}, {"row": "3",
   "col": "2", "val": "0", "selected": true}, {"row": "3", "col": "3", "val": "0", "
  selected": true}, {"row": "3", "col": "4", "val": "0", "selected": true}, {"row": "4",
   "col": "0", "val": "0", "selected": true}, {"row": "4", "col": "1", "val": "0", "
  selected": true}, {"row": "4", "col": "2", "val": "0", "selected": true}, {"row": "4",
   "col": "3", "val": "0", "selected": true}, {"row": "4", "col": "4", "val": "0", "
  selected": true}]}, "grid": [[0, 0, 0, 0, 0], [0, 0, 0, 0, 0], [0, 0, 0, 0, 0], [0, "2
  ", 0, "2", 0], ["2", 0, "2", 0, "2"]], "currentLayer": 0, "layer_list": [[[0, 0, 0, 0,
   0], [0, 0, 0, 0, 0], [0, 0, 0, 0, 0], [0, "2", 0, "2", 0], ["2", 0, "2", 0, "2"]]], "
  submit": 1, "time": 3811}]}"
6 }
```

*Listing 1.* Example of the human trace in JSON file

## B. Tackled ARC and Mini-ARC Problems

In this section, we describe ARC problems that we address and show with figures. Figure 9 presents a selection of ARC problems addressed using our model. The name of each problems is given by us for distinguishing and demonstrating easily.

- *Diagonal Flip*: This problem operates on the rule that the shape of an object is transformed according to a specific *diagonal flip* direction. (Mini-ARC; Task ID: l6abdiipodvgey6tbdf, l6ad1nnu454mki54lqa)

- *Tetris*: This problem adheres to the rule that objects descend in a manner reminiscent of the gameplay in Tetris. (Mini-ARC; Task ID: l6ab7fu64lvutswrtbk)

- *Gravity*: This problem functions under the rule that objects attach to a certain designated object within the task. (ARC; Task ID: 4093f84a)

- *Stretch*: This problem is about moving the given object to the top and extending the pixels that were at the bottom. (Mini-ARC; Task ID: l6acmlt1nkjxwh68ah)

## C. Categorize Object-Centric ARC problem

Prior to designing the PnP algorithm, we embarked on a preliminary study to understand how objects are defined in the ARC problem and what a priori knowledge is leveraged when humans attempt to solve ARC problems. An insightful study by Moskvichev et al. (2023) intuitively deconstructs the ARC problem into elements such as copying, counting, object extraction, and moving to a boundary. However, our focus was predominantly on the use of objects in the ARC problem, leading us to categorize the original ARC problem (Chollet, 2019) into either object-centric or not.

As briefly introduced in Section 3.3, we conducted an analysis of 400 ARC learning problems and singled out 128 object-centric problems. These selected problems were subsequently classified based on four specific attributes: (1) Definition of Object - Color, (2) Definition of Object - Shape, (3) Method of Object Modification, and (4) Consideration of Input/Output Simultaneously.

*(1) Definition of Object - Color* attribute indicates whether an object in an ARC problem is composed of pixels of the same color. If each element shares the same color property, the term 'same color' is assigned. Conversely, 'mixed color' is used when the object comprises two or more colors.

*(2) Definition of Object - Shape* refers to the classification based on the pixel relationships within an object. There are four possible values for this attribute, representing whether pixels: share a common boundary or are directly adjacent, share a single common corner or are diagonally adjacent, are organized within a specific range, or overlap within multiple objects.

*(3) Method of Object Modification* attribute pertains to the way an object is modified during problem-solving. Some problems may involve altering the object in the output based on a specific rule. This category includes operations such as copying, coloring, moving, selecting, and counting.

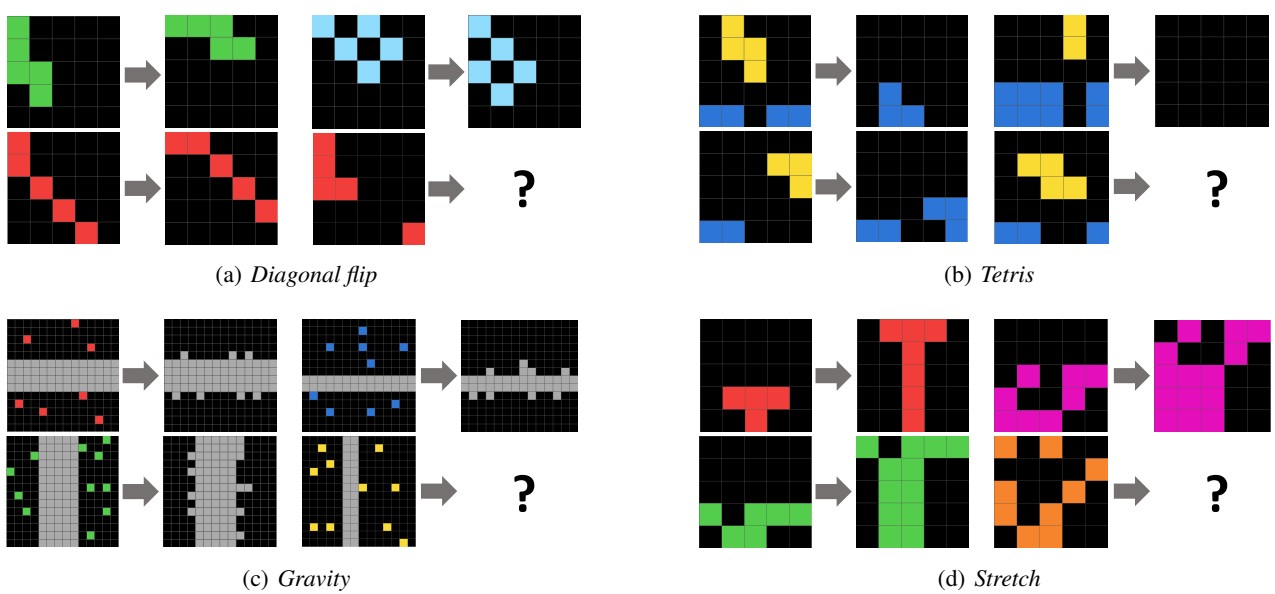

(a) *Diagonal flip*

(b) *Tetris*

(c) *Gravity*

(d) *Stretch*

*Figure 9.* ARC and Mini-ARC problems tackled in this paper

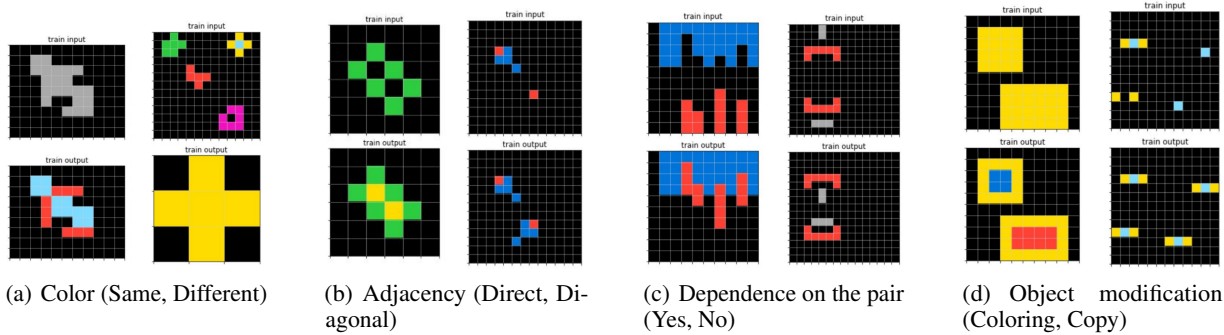

(a) Color (Same, Different)

(b) Adjacency (Direct, Diagonal)

(c) Dependence on the pair (Yes, No)

(d) Object modification (Coloring, Copy)

*Figure 10.* Category of the ARC problem based on four specific attributes

Lastly, *(4) Consideration of Input/Output Simultaneously* is an attribute reflecting whether both input and output must be considered concurrently when detecting and defining an object. Given the characteristics of the ARC problem, this attribute was included as the definition of an object could vary depending on the output, even with the same input.

## D. Performance of the PnP algorithm

Following the brief description in 4.1.3 we provide details on the object detection performance of the PnP algorithm. We calculated recall scores for measuring the performance of detecting objects for each ARC problem and used silhouette scores to evaluate the quality of clustering. The recall score for each ARC problem is calculated by Equation (3).

$$\text{Recall} = \frac{\text{Number of correctly detected objects}}{\text{Total number of observable objects in the problem}} \tag{3}$$

The silhouette score measures the performance of the clustering algorithms by calculating the cohesion and separation of data points. It ranges from -1 to 1, with a smaller coefficient indicating that an object was included in the wrong cluster and a larger number indicating that the object was correctly assigned to its own cluster. For data point i, the silhouette coefficient is calculated as follows:

$$s = \frac{b_i - a_i}{\max(b_i, a_i)}, \tag{4}$$

where $b_i$ represents the distance within the cluster, or the average distance to the nearest cluster, and $a_i$ represents the distance outside the cluster or the average distance to other clusters. If the silhouette factor is greater than 0.7, the structure of the cluster can be considered accurate.

*Table 3.* Evaluating Push and pull algorithm result: Clustering performance

| Category | recall | silhouette score |
|---|---|---|
| Same color, Direct adjacency | 0.96 | 0.63 |
| Same color, Diagonal adjacency | 0.81 | 0.54 |
| Different color, Direct adjacency | 0.81 | 0.54 |
| Different color, Diagonal adjacency | 0.62 | 0.40 |
| Same color, Overlap | 0.65 | 0.48 |
| Same color, With in specific range | 0.80 | 0.59 |
| Overall | 0.88 | 0.57 |

Table 3 represents the recall value and silhouette score for 128 object-oriented problems according to each category. As shown in the table, except in two cases; Different colors with Diagonal adjacency and the same color with overlaps, the PnP algorithm detects objects well with performance over 80 percent. In particular, in the case of problems with the same color with direct adjacency which includes the most number of problems, the correct answer rate was 96 percent. The silhouette score also has a similar pattern to the recall value, and when the pixels that form the object are directly adjacent the silhouette score tends to increase as they are clearly clustered to each other.

In addition to functionality, we also address the efficiency of the PnP algorithm. The PnP algorithm requires two arrays: one for the number of nodes times the number of attributes, and another for the number of edges. Given that it performs a graph abstraction first, the maximum space complexity is $30 * 30 * 4 = 3600$.

Regarding time complexity, it encompasses two primary operations: push and pull and cluster assignment, which is implemented via DBSCAN. In the push and pull operation, the algorithm inspects nodes containing color information to determine distances. This results in a time complexity of $O(N^2)$, with N being the maximum number of pixels (900 in this case). DBSCAN's operation is recognized to have a time complexity of $O(N \log N)$. Consequently, the overall time complexity of our algorithm is dominated by the push and pull operation, culminating in a complexity of $O(N^2)$. Nevertheless, given that the maximum size in the ARC problem is sparse, the algorithm operates relatively effectively.

## E. Decision Transformer

### E.1. Establishing Reward Function for the Decision Transformer

Prior to exploring specific methodologies, it is important to consider the role of rewards in training the Decision Transformer model. Reward functions guide the model towards desirable states and actions, providing valuable feedback during the learning process. The design of these functions plays a critical role in the learning efficiency and overall performance of the model. The following strategies illustrate different approaches we have considered for setting up the reward function in our experiment.

1. (Equally Distributed Reward) In the experiment, we contemplated how to design the rewards in order to incorporate return-to-go into the Decision Transformer. For this experiment, we set the starting state as 0 and the final state as 1, and divided the return-to-go into equidistant intervals based on the time step. When training the model with equidistant return-to-go, the model is compelled to predict return-to-go at the same intervals during the prediction process. This aspect can act as a disadvantage during the training process of the Decision Transformer. However, when return-to-go is used in training, it provides additional information for predicting the next step, which can lead to performance improvement. The influence of return-to-go varies depending on the type of problem and the length of the trace.

Therefore, it is important to consider the pros and cons when using return-to-go, and apply it appropriately when its benefits outweigh its drawbacks. In problems where the influence of return-to-go is small, the DT without return-to-go (BC; Behavioral Cloning) may show better results than the DT with return-to-go (Vanilla DT). In fact, in the Tetris and Gravity problems, the DT without return-to-go showed better performance, while in the Diagonal_flip and Stretch problems, the DT with return-to-go demonstrated better performance. Therefore, if a more suitable return-to-go is used for Decision Transformer, it could potentially show better performance than equidistant return-to-go.

2. (Weighted Reward) How should we define the rewards in our reinforcement learning model? We consider a situation where we have an abundance of ARC-traces that have solved a given problem. These traces can be represented as a state-space graph, where each node corresponds to a grid state from the traces. Given a sufficiently large state-space graph, we can define key states as those most frequently reached by users. From our current state, we can utilize conditional probabilities to identify actions that are likely to lead us to these key states. These actions are likely important, and we propose that reward values could be distributed differentially based on their significance. For example, in the context of the Tetris problem, a high reward could be assigned to the action of clearing a fully filled line, an event that is inevitable in the game progression. In addition, considering that we are currently utilizing around ten different DSLs, the number of states accessible before reaching the final state is limited. As a potential solution, we propose a method of back-tracking from the final state to prepare a set of states reachable within one or two hops. Providing a high reward for reaching these states could potentially increase the efficiency of model learning.

### E.2. Contemplating Improved Experimental Setups

The ARC challenge presents a peculiar issue: the patterns within training and evaluation data differ significantly, which implies a model trained on the former cannot readily generalize to the latter. Consequently, we risk criticism for circumventing the essence of ARC by adopting a purely supervised learning experimental setup. Therefore, there may be doubts about the results from our current experiments, and here are some concerns.

**Question:** We augmented the data using actions from expert traces. Given that we provided all the augmented ARC traces, which include both action and state sequences as training data, we should be capable of discovering viable solutions for all input grids by merely using the raw expert traces. We even incorporated the sophisticated Decision Transformer algorithm; why did we fail to reach 100% performance?

**Answer:** We posit that if only a single expert trace were included, we might have achieved 100% performance. However, as we augmented the data using various types of expert traces, there exist diverse combinations and possibilities of states and actions during the learning process. For example, in a particular test state, the correct course of action may be to 'rotate followed by 'reflect_y'. However, within our existing traces, the frequency of sequences where we 'rotate' and then 'reflect_x' might be higher. This discrepancy could prevent a quick solution from mimicking the action sequence of the correct trace, thus hindering 100% accuracy. Consequently, the accuracy may be lower than expected.

So, what might be a better experimental setting?

1. While this setup does not include the core concepts of the ARC problem, namely open-set and few-shot, I propose the following idea. Should we have trained the Decision Transformer model with all augmented datasets (40,000) corresponding to four problems as input data, and then given a test example randomly pertaining to one of the four problems, and had it solved? → If we were to conduct an experiment in such a setup, we could solve the problem by first determining which problem the test case (+ 3-shot pair) is, using a neural architecture search module placed on each individual Decision Transformer, and then learning the Decision Transformer that solves that problem. The maximum performance obtainable in that case would be the performance reported in the Table 2.

2. A more essential approach to the ARC is as follows: → We should use the model trained with problems A, B, C, and D to solve new problems E, F, G that we have never encountered before. As we proceed with our research, experiments must definitely be conducted in this setup.

## F. Example of expert traces

In Appendix F, we define expert traces for the *diagonal flip* problem. These traces guarantee to provide the correct solution without any detours. The corresponding expert traces are then utilized for data augmentation purposes. The listings below depict these expert traces for the *diagonal flip* problem.

```
1  seq_action = [
2      ["start", "reflectx", "rotate", "end"],
3      ["start", "rotate", "rotate", "reflecty", "rotate", "end"],
4      ["start", "reflectx", "reflecty", "rotate", "reflectx", "end"],
5      ["start", "rotate", "rotate", "rotate", "reflectx", "end"],
6      ["start", "rotate", "reflectx", "rotate", "rotate", "end"],
7      ["start", "reflecty", "reflectx", "rotate", "reflectx", "end"],
8      ["start", "rotate", "rotate", "reflectx", "rotate", "rotate", "rotate", "end"],
9      ["start", "reflecty", "rotate", "rotate", "rotate", "end"],
10     ["start", "reflectx", "reflectx", "rotate", "reflecty", "end"],
11     ["start", "rotate", "reflecty", "end"], #TRACE_3431
12 ]
```

*Listing 2.* Examples of expert traces used to create an augmentation of the Diagonal flip task

In Listing 2, all traces can be classified as expert traces as they contain no unnecessary or inefficient actions. Each trace is less than the defined threshold in length, contains no `edit` actions, and has no detectable cycles among the actions. Thus, they are all categorized as expert traces.

## G. Algorithm details

In this section, we will describe the models and training methods in more detail.

### G.1. Data Augmentation Algorithm

---

**Algorithm 1** Data Augmentation

---

**Input:** a list of human traces($tr\_in$)
**Output:** a list of augmented traces($tr\_aug$)

**def** `IsExpertTrace`($trace$)**:**
    **if** *(len(trace) $\geq$ threshold)* **or** *("edit" $\in$ trace)* **or** *(some cycles are found in trace)* **then**
        ∟ **return** False
    **return** True

**def** `GenerateTrace`($grid, exp\_trace$)**:**
    $trace \leftarrow [grid]$
    **for** $action \in exp\_trace$ **do**
        $grid \leftarrow$ a new grid updated by applying $action$ to current $grid$.
        $trace \leftarrow trace + [grid]$
    **return** $trace$

/* Select expert traces. */
$tr\_exp \leftarrow []$
**for** $trace \in tr\_in$ **do**
    **if** `IsExpertTrace`*(trace)* **then**
        $tr\_exp \leftarrow tr\_exp + [trace]$

/* Generate synthetic traces. */
$tr\_aug \leftarrow []$
**for** $1 \leq i \leq 10000$ **do**
    $grid \leftarrow$ A 2D array whose size is $N \times N$ and each element is one of limited values.
    **for** $exp\_trace \in tr\_exp$ **do**
        $aug\_trace \leftarrow$ `GenerateTrace`($grid, exp\_trace$)
        $tr\_aug \leftarrow tr\_aug + [aug\_trace]$

**return** $tr\_aug$

---

Algorithm 1 encapsulates the data augmentation method outlined in Section 3.1.2. The algorithm begins by defining expert traces, identified as trajectories collected via O2ARC (Kim et al., 2022). These represent the strategies employed by humans to solve Mini-ARC problems, devoid of unnecessary or inefficient steps. We removed unnecessary steps by imposing a maximum length of $threshold$ on the traces, as longer traces tend to incorporate superfluous steps. In this context, we strategically adjust the threshold value to suit the specific requirements of each problem in Mini-ARC. For instance, in the case of the *diagonal flip* problem, the threshold is set to 6. In this process, we discarded traces containing the `edit` action, which alters the color of a single pixel individually, as well as traces that exhibited cyclical patterns. In more detail, the `edit` action is inefficient and reduces the generality of solutions, thus we exclude all traces employing it from the expert traces. Cycles, defined as unnecessary sequences where two or more consecutive actions lead back to the initial state, were also eliminated from the traces.

Subsequently, we generate an appropriate 2D random grid tailored to the type of the current Mini-ARC problem. The specific coloring and layout of the grid are determined by the problem type. The size of grid is set to $N \times N$, where N is primarily set to 5 but adjusted to 7 exclusively for the *gravity* problem. *Diagonal flip* and *stretch* problems use black and one other randomly chosen color, while *tetris* and the *gravity* problems employ black and two randomly chosen colors. The layout of colored pixels (non-black) also varies according to the problem type. For *diagonal flip* and *stretch* problems, colored pixels are randomly distributed, whereas the *tetris* problem utilizes pixel placement that mirrors the shape of blocks from the Tetris game. The *gravity* problem distinctively features a central line composed of same-colored pixels. This process yields a total of $10,000$ instances.

Lastly, we detail the data augmentation process for creating a new set of traces using the set of expert traces and the 2D random grids. We introduce the function `GenerateTrace`, which creates a single trace by applying the actions from an expert trace sequentially to a 2D random grid. Every time an action is applied, the grid updates, and the sequence of these updated grids forms the new trace. In essence, the generated trace embodies the process of applying an expert trace, representing an efficient solution to a specific Mini-ARC problem, onto a different random grid. Algorithm 1 accumulates the results of performing `GenerateTrace` function for each expert trace across $10,000$ instances, culminating in the final output.

### G.2. Push and Pull Clustering Algorithm

Algorithm 2 elaborates on the object detection process, known as the Push and Pull (PnP) clustering algorithm, as discussed in Section 3.2. To be more precise, it encapsulates the graph abstraction operation explained in Section 3.2.1, the Push and Pull operation described in Section 3.2.2, and the cluster assign operation outlined in Section 3.2.3. These processes are respectively represented by $f$, $g$, and the DBSCAN in Figure 4.

Examining the graph abstraction in greater detail, function $f$ accepts a 2D grid as input and constructs a graph, where each pixel in the grid is a node, and edges are drawn between directly or diagonally adjacent nodes. In this construction, each node stores the coordinate and color information of its corresponding pixel. Each edge, on the other hand, holds a value between 1 and 5, which depends on the coordinate and color information of its connecting nodes. For instance, an edge connecting two directly adjacent nodes of the same color is assigned a value of 1, whereas an edge between two diagonally adjacent nodes of different colors receives a value of 5. In Algorithm 2, the function `CalculateDistance` is utilized to compute the value to be stored on an edge between two adjacent pixels. Furthermore, in the portion of Algorithm 2 describing function $f$, $pixel_i$ (a non-black pixel in the grid) and $pixel_j$ (a non-black pixel among the eight pixels adjacent to $pixel_i$) are selected by the nested loops. The value to be stored on the edge between the selected $pixel_i$ and $pixel_j$ is computed using `CalculateDistance`, and this edge is added to the $edges$ set.

Function $g$ operates on the graph constructed through the previously described processes, with particular regard to the values stored on each edge. When an edge harbors a value of 4 or 5, it signifies a repulsive force at work between the two linked nodes. This force is instrumental in moving the nodes away from each other during the subsequent rearrangement process. Conversely, an edge holding a value of 1 or 2 is interpreted as exerting an attractive force between the connected nodes, causing them to gravitate towards each other during the rearrangement. This procedure extends to include an examination of all edges, which in turn influences the relocation of the nodes based on the observed forces. Function $g$ meticulously regulates this process, adjusting the placement of nodes according to the inferred forces from the edge values, thereby ensuring the optimal organization of the graph.

The final part of Algorithm 2 delineates the process of assigning a unique cluster number to each pixel in the grid. This is accomplished by implementing the DBSCAN algorithm to conduct the push and pull operation on the grid, which has

---

**Algorithm 2** Push and Pull Clustering

---

**Input:** a 2D array of ARC problem($grid$)
**Output:** a 2D array of ARC problem with assigned cluster number($grid$)

**def** ManhattanDistance($pixel_i, pixel_j$)**:**
    **return** ABS($pixel_i.x$ - $pixel_j.x$) + ABS($pixel_i.y$ - $pixel_j.y$)

**def** CalculateDistance($pixel_i, pixel_j$)**:**
    $distance \leftarrow 1$
    **if** $pixel_i.color \neq pixel_j.color$ **then**
        $distance \leftarrow distance + 3$
    **if** ManhattanDistance($pixel_i, pixel_j$) == 2 **then**
        $distance \leftarrow distance + 1$
    **return** $distance$

/* Function $f$ ; Graph abstract operation. */
$edges \leftarrow []$
**for** $pixel_i \in \{pixel \,|\, pixel \in grid$ **and** $pixel.color \neq$ "black"$\}$ **do**
    **for** $pixel_j \in \{pixel \,|\, pixel \in grid$ **and** ManhattanDistance($pixel, pixel_i$) == 1 **and** $pixel.color \neq$ "black"$\}$
    **do**
        $distance \leftarrow$ CalculateDistance($pixel_i, pixel_j$)
        $edges \leftarrow edges + [(pixel_i, pixel_j, distance)]$

/* Function $g$ ; Push and pull operation. */
**for** $edge \in edges$ **do**
    $repulsion \leftarrow (edge.distance - 3)/2$
    **if** $repulsion \geq 0$ **then**
        Push $edge.pixel_i$ and $edge.pixel_j$ in $grid$ based on $repulsion$.
    **else**
        Pull $edge.pixel_i$ and $edge.pixel_j$ in $grid$ based on $repulsion$.

/* DBSCAN ; Assigning cluster operation. */
$clusters \leftarrow$ DBSCAN($grids$)
$pixel\_order \leftarrow 1$
**for** $cluster \in clusters$ **do**
    **for** $node \in cluster$ **do**
        **if** $node.color \neq$ "black" **then**
            $pixel \leftarrow grid[node.x][node.y]$
            $pixel \leftarrow pixel + [pixel\_order]$
    $pixel\_order \leftarrow pixel\_order + 1$
**return** $grid$

---

undergone position adjustments. Following this, each cluster identified by DBSCAN is assigned a unique number, and this cluster identification is then integrated into every pixel. Upon completion of these steps, the modified grid is returned, signifying the completion of Algorithm 2.

**G.3. Decision Transformer with Push and Pull Clustering Algorithm**

Algorithm 3 illustrates the procedure of resolving the Mini-ARC problem utilizing the Decision Transformer, which in this research, has been supplemented with object detection information. This algorithm is essentially a derivative of Algorithm 1 from the original Decision Transformer (Chen et al., 2021), albeit with a few adaptations. It includes some additional steps to facilitate object detection and modifications specifically designed to handle the Mini-ARC problem. However, it's important to note that the major part of the process remains consistent with the original Decision Transformer algorithm. Thus, considering the high degree of overlap and in the interest of brevity, we have decided not to include a step-by-step, detailed discourse of the entire process within the scope of this paper. Instead, we will focus on the novel additions and modifications that contribute to the unique aspects of our proposed methodology.

In the `DecisionTransformer` function, modifications have been implemented to incorporate object detection information. To begin with, a new variable, $p\_embedding$, and a linear embedding layer called embed_p have been introduced. Subsequently, in the construction of $input\_embeds$, $p\_embedding$ is added to the previously used $s\_embedding$, $a\_embedding$, and $r\_embedding$. This addition of $p\_embedding$ equips the `DecisionTransformer` with the ability to access pixel cluster information at each state during the learning process. As a result, the `DecisionTransformer` is now able to perceive pixels that share the same cluster number as a single object. In terms of return values, instead of providing a single action-related predicted value, the revised function now returns three predicted values: state, action, and return-to-go. This revision was necessitated by a learning issue in the ARC task where returning only the action proved to be insufficient. As the same issue was found in the Mini-ARC, the function was adapted to return state, action, and return-to-go, thereby addressing this problem.

The main changes during the model's training process are reflected in the return value changes in the `DecisionTransformer` function and the corresponding alterations to the loss function. The original Decision Transformer assumed that the return value, action, was continuous, and hence used the Mean Squared Error (MSE) function for it. However, in this research, not only is action added to the return values but also state and return-to-go. Moreover, in the Mini-ARC problem, the action and state are discrete, while only return-to-go is continuous. Therefore, the loss function in this research is determined as the sum of the cross-entropy function for action and state, and the MSE function for return-to-go.

Changes during the testing phase also stem from the expanded return values of the `DecisionTransformer` function. Upon examining the loop, it's evident that the original Decision Transformer only returns a prediction for the action, necessitating further steps to compute the state and return-to-go. Contrarily, our version of the `DecisionTransformer` returns predictions for all three parameters, thus eliminating these extra steps.

**Algorithm 3** Decision Transformer with Push and Pull Clustering

---

**Input:** state($s$), action($a$), return-to-go($r$), timestep($t$)
**Output:** predicted final state($final\_s$)
```
/* transformer ; a transformer with causal masking (GPT) */
/* embed_t ; a learned episode positional embedding layer */
/* embed_s, embed_a, embed_r, embed_p ; embedding layers */
/* pred_s, pred_a, pred_r ; linear prediction layers */
```
**def** DecisionTransformer($s, a, r, t$)**:**
  $pos\_embedding \leftarrow$ embed_t($t$)
  $s\_embedding \leftarrow$ embed_s($s$) $+ pos\_embedding$
  $a\_embedding \leftarrow$ embed_a($a$) $+ pos\_embedding$
  $r\_embedding \leftarrow$ embed_r($r$) $+ pos\_embedding$
  $p\_embedding \leftarrow$ embed_p(Push_and_Pull_Clustering($s$)) $+ pos\_embedding$

  $input\_embeds \leftarrow (s\_embedding, a\_embedding, r\_embedding, p\_embedding)$
  $hidden\_states \leftarrow$ transformer($input\_embeds$)

  $new\_s \leftarrow$ pred_s($hidden\_states.s$)
  $new\_a \leftarrow$ pred_a($hidden\_states.a$)
  $new\_r \leftarrow$ pred_r($hidden\_states.r$)
  **return** ($new\_s, new\_a, new\_r$)

```
/* Train the Decision Transformer */
```
**for** $(s, a, r, t) \in Train\_Dataloader$ **do**
  $(pred\_s, pred\_a, pred\_r) \leftarrow$ DecisionTransformer($s, a, r, t$)
  $loss \leftarrow$ Cross_Entropy($(pred\_s, pred\_a), (s, a)$)$+$ MSE($pred\_r, r$)
  optimizer.zero_grad()
  $loss$.backward()
  optimizer.step()

```
/* Test the Decision Transformer */
```
$K \leftarrow 5$
$(s, a, r, t) \leftarrow (initial\_state, \text{``start''}, 0.0, 1)$
**while** $true$ **do**
  $new\_s, new\_a, new\_r \leftarrow$ DecisionTransformer($s, a, r, t$)$[-1]$
  **if** $new\_a ==$ "end" **then**
    $final\_s = new\_s$ break
  $(s, a, r, t) \leftarrow (s + [new\_s], a + [new\_a], r + [new\_r], t + [len(r) + 1])$
  $(s, a, r, t) \leftarrow (s[-K :], a[-K :], r[-K :], t[-K :])$

**return** $final\_s$

---