# OpenReview forum: "Unraveling the ARC Puzzle: Mimicking Human Solutions with Object-Centric Decision Transformer"
_ICML.cc/2023/Workshop/ILHF — ILHF Workshop ICML 2023_

### Official Review · Reviewer_2ceT · 2023-06-13
**Interesting work; bold and unclear claims**

**Rating:** 6
**Confidence:** 4

**Review:**

This work aims to solve ARC problems using object-centric imitation learning. In particular, it detects objects using a clustering method and trains a decision transformer to mimic human solutions.

The method is technically sound. Intuitively, it makes sense to learn object-centric policies to solve ARC tasks.

However, it is unclear how solving ARC is connected to achieving AGI. Why would progress on ARC indicates progress on AGI? How would the proposed method help solve a broader range of problems? The current submission only discusses how to solve ARC but does not provide a broader discussion on the implication of the study.

---

### Official Review · Reviewer_LaLD · 2023-06-16
**Review of Paper 28**

**Rating:** 7
**Confidence:** 3

**Review:**

**Summary of Contributions**

The authors propose an imitation learning approach combined with decision transforms for solving the ARC puzzle. In addition to passing the raw states to the decision transformer, they augment it with the PnP algorithm to form object clusters. By incorporating this additional feature extraction as an input, the authors demonstrate that the accuracy of ARC tasks improves compared to different baselines, namely behavior cloning with and without PnP, and decision transformer without PnP.

**Strengths and Weaknesses**

+ Application of decision transforms and augmentation with additional feature extraction steps.
+ Emphasis on the importance of object detection and its impact on accuracy.
+ Detailed evaluation and explanation of the methodology, particularly in the appendix.

Weaknesses and Questions

- Can this feature extraction step be integrated into the learning process itself? Is there a reason why the decision transformer cannot infer these features directly from the state information?
- Table 2 shows that behavior cloning (+PnP) outperforms the decision transformer (+PnP). Is there a specific reason for this? It is also unclear whether the PnP algorithm or the decision transformer has a greater influence on the results. What happens when the values in Table 1 are changed, learned, or fine-tuned? Does that have any impact?

Minor (Nitpicks)

The abstract is quite generic. It would be beneficial if the authors provided more information about the methods and results from the paper.
The results section primarily refers to figures and tables without providing explicit inferences or explanations. It would be helpful if the authors could draw inferences from the results or provide further explanations.

---

### Decision · Program_Chairs · 2023-06-20

Accept